# High-Fat, Western-Style Diet, Systemic Inflammation, and Gut Microbiota: A Narrative Review

**DOI:** 10.3390/cells10113164

**Published:** 2021-11-14

**Authors:** Ida Judyta Malesza, Michał Malesza, Jarosław Walkowiak, Nadiar Mussin, Dariusz Walkowiak, Raisa Aringazina, Joanna Bartkowiak-Wieczorek, Edyta Mądry

**Affiliations:** 1Department of Pediatric Gastroenterology and Metabolic Diseases, Poznan University of Medical Sciences, 61-701 Poznań, Poland; ida.malesza@ump.edu.pl (I.J.M.); jarwalk@ump.edu.pl (J.W.); 2Department of Physiology, Poznan University of Medical Sciences, 61-701 Poznań, Poland; mmalesza6@gmail.com (M.M.); joanna@wieczorek.net.pl (J.B.-W.); 3Department of General Surgery, West Kazakhstan Marat Ospanov Medical University, Aktobe 030012, Kazakhstan; nadiarm2001@mail.ru; 4Department of Organization and Management in Health Care, Poznan University of Medical Sciences, 61-701 Poznań, Poland; dariuszwalkowiak@ump.edu.pl; 5Department of Internal Diseases No. 1, West Kazakhstan Marat Ospanov Medical University, Aktobe 030012, Kazakhstan; raisa_aringazina@mail.ru

**Keywords:** postprandial inflammation, endotoxemia, TLR4, NF-κB, dysbiosis, leaky gut, LPS, bile acids, oxidative stress, endoplasmic reticulum stress

## Abstract

The gut microbiota is responsible for recovering energy from food, providing hosts with vitamins, and providing a barrier function against exogenous pathogens. In addition, it is involved in maintaining the integrity of the intestinal epithelial barrier, crucial for the functional maturation of the gut immune system. The Western diet (WD)—an unhealthy diet with high consumption of fats—can be broadly characterized by overeating, frequent snacking, and a prolonged postprandial state. The term WD is commonly known and intuitively understood. However, the strict digital expression of nutrient ratios is not precisely defined. Based on the US data for 1908–1989, the calory intake available from fats increased from 32% to 45%. Besides the metabolic aspects (hyperinsulinemia, insulin resistance, dyslipidemia, sympathetic nervous system and renin-angiotensin system overstimulation, and oxidative stress), the consequences of excessive fat consumption (high-fat diet—HFD) comprise dysbiosis, gut barrier dysfunction, increased intestinal permeability, and leakage of toxic bacterial metabolites into the circulation. These can strongly contribute to the development of low-grade systemic inflammation. This narrative review highlights the most important recent advances linking HFD-driven dysbiosis and HFD-related inflammation, presents the pathomechanisms for these phenomena, and examines the possible causative relationship between pro-inflammatory status and gut microbiota changes.

## 1. Introduction

The Western diet (WD) is a commonly used term but lacks a precise definition. It can be broadly characterized by overeating and frequent snacking, leading to a prolonged postprandial state [1,2]. The process of dietary and lifestyle changes began after the Neolithic Revolution and was related to the development of agriculture, animal domestication, and technological progress. These factors significantly lowered the cost of food and increased availability [1]. Subsequently, the caloric intake increased beyond energy expenditure with the industrial revolution about 250 years ago, a unique event in human history [2]. Moreover, a calorically rich WD combined with chronic overeating is closely related to the Western lifestyle (e.g., physical inactivity, avoidance of sun exposure, insufficient sleep, increased chronic psychological stress, smoking, and environmental pollution) [2,3]. These described changes are very recent from an evolutionary perspective and have not allowed the human genome to adapt. Thus, the mismatch between human physiology and WD and lifestyle, combined with a prolonged lifespan has contributed to the development of so-called civilization diseases [1,2].

Interestingly, almost everyone has an intuitive understanding of the WD concept, but the strict digital expression of nutrient ratios requires further investigation. Generally, the WD is characterized by high intakes of refined sugars (candies and sweets, and high-sugar soft drinks), animal fats (high intake of saturated and omega-6 fatty acids, reduced omega-3 fat intake), processed meats (especially red meat), refined grains, high-fat dairy products, conventionally-raised animal products, salt, eggs, potatoes, corn [4,5], mainly processed, refined, fried, and pre-packaged, with low intakes of unprocessed fruits, vegetables, whole grains, grass-fed animal products, fish, nuts, and seeds, hence, is low in fiber, vitamins, minerals, and other plant-derived molecules such as antioxidants [1,5,6,7,8,9]. US data for 1908–1985 on the proportion of macronutrients consumed showed that the percentage of total calories available from carbohydrates decreased from 57% to 46%, whereas the total calories available from fats increased from 32% to 43%. In the following years, the trend for macronutrient consumption was maintained, with carbohydrate intake decreasing to about 35%, while the consumption of fats increased to 45% [10]. Although the data was released in 1989, this allows for understanding fat and carbohydrates intake trends in modern society.

Besides the quantitative problem of macronutrients consumed in the WD, their quality is also unfavorable. The high consumption of simple carbohydrates, mainly those previously refined, contributes to the WD’s harmful metabolic properties. Additionally, it is associated with low consumption of high-quality carbohydrates from fruits, vegetables, beans, legumes, and whole grains, which are sources of essential vitamins, minerals, and nutrients [1,7,8,10]. The high intake of saturated fats, including excessive amounts of omega-6 polyunsaturated fatty acids (PUFA), small amounts of omega-3 PUFA, and an unhealthy omega-6/omega-3 ratio of 20:1 is especially harmful in terms of the metabolic consequences [7,11,12]. The WD can be considered as an unhealthy diet with high consumption of fats, however, as the macronutrient composition of the WD is not precisely defined, various fat percentages are assumed as standards in different studies, as shown in Table 1.

Other essential components of WD lately being carefully scrutinized for their purported adverse effects are food additives like emulsifiers or sweeteners. Widely used in fat-based foods, emulsifiers are proven to impair gut barrier function [13]. They also stimulate low-grade inflammation, contributing to obesity and metabolic syndrome in wild-type hosts or even developing robust colitis in murine models predisposed to this disorder [14]. Moreover, consumption of emulsifiers is positively correlated with both obesity and metabolic syndrome in humans [15].

Permanent overeating promotes multiple civilization diseases related to several somatic and mental issues caused by metabolic disturbances, the most important of which are hyperinsulinemia, insulin resistance, dyslipidemia, overstimulation of the sympathetic nervous system, low-grade systemic inflammation, renin-angiotensin system (RAS) overactivation, dysbiosis, endotoxemia, increased production of reactive oxygen species (ROS), and oxidative stress (OxS) (Figure 1) [16,17,18,19,20,21,22,23,24,25]. These pathophysiologies have been linked to diseases like obesity, type 2 diabetes mellitus, dyslipidemia, inflammatory bowel disease, neoplasms, and cardiovascular diseases (including atherosclerosis, cardiomyopathy, hypertension, and heart failure) [26,27,28]. Furthermore, the WD promotes alterations in gut microflora, resulting in dysbiosis, gut barrier dysfunction, increased intestinal permeability, and leakage of toxic bacterial metabolites into the circulation, all of which contribute to the development of low-grade systemic inflammation [29,30,31,32,33,34,35]. This phenomenon is also considered to be associated with inflammation-related ageing of organisms called “inflammaging”. This topic is thoroughly discussed by Calder et al. review [36].

This review highlights the most important recent advances linking HFD-driven dysbiosis and HFD-related inflammation, presenting the pathomechanisms for these phenomena, and examines the possible causative relationship between proinflammatory milieu and gut microbiota changes.
cells-10-03164-t001_Table 1Table 1High-fat diets.ReferenceStudy ModelsFat (%kcal)Cani et al., 2008 [37]C57bl6/J mice and ob/ob mice C57bl6 background72%Garidou et al., 2015 [38]Wildtype (WT) C57Bl6/J mice, (RORγt−/−) mice, Rag1-deficient (Rag1−/−) mice, OVA-specific TCR transgenic (OTII) mice72%Tomas et al., 2016 [39]mice (C57BL/6JRj, Janvier, France)71%Agus et al., 2016 [29]C57BL/6 mice60%Amar et al., 2011 [40]C57bl6, ob/ob, CD14/, ob/obxCD14/, Myd88/, Nod1/or Nod2/mice60%Brandsma et al., 2019 [41]Female Casp1−/− mice (B6N.129S2-Casp1tm1Flv/J)) and Ldlr−/− mice (B6.129S7-Ldlrtm1Her/J)60%Chelakkot et al., 2018 [42]C57BL/6 mice60%Crawford et al., 2019 [30]Sprague-Dawley rats60%Guo et al., 2017 [43]C57BL/6 mice60%Hu, Zhang, 2016 [44]Toll-like receptor 4 knockout (TLR4−/−) and C57BL/6J (WT) mice60%Jeong et al., 2019 [45]Male C57BL/6 J mice60%Kawano et al., 2016 [46](M-Ccr2KO) and (Vil-Ccl2KO) mice60%Kim et al., 2012 [47]C57BL/6J and TLR4-deficient C57BL/10ScNJ mice60%In Kim et al., 2019 [48]C57BL/6 mice60%Li et al., 2019 [49]C57BL/6 mice60%Perez et al., 2019 [50]C57BL/6 mice (IL-17RA−/−)60%Schmid et al., 2015 [51]Healthy human60%Talukdar et al., 2012 [52]NE KO, JAX labs B6.129X1–Elanetm1Sds/J mice and WT C57BL/6J mice60%Wang et al., 2020 [53]C57BL/6J mice60%Gulhane et al., 2016 [54]Wild type (WT) C57BL/6 mice46%de la Serre et al., 2010 [55]Male Sprague Dawley rats45%Kim et al., 2019 [31]male C57BL/6 J mice45%Park et al., 2016 [56]ApcMin/+ mice45%Sen et al., 2017 [57]Male Sprague Dawley rats45%Napier et al., 2019 [58]BALB/c mice, C57BL/6 mice42%Murakami et al., 2016 [59]C57/BL6 mice40%Wan et al., 2019 [60]Healthy adults40%Laugerette et al., 2011 [61]C57Bl6/J mice37.7%Guo et al., 2016 [62]C57BL/6J ApoE−/− mice37%


## 2. Methodology

PubMed/Medline and Cochrane databases were searched for studies about the topic of interest. The following English terms and their combinations were used: high-fat diet, microbiota, low-grade inflammation, inflammation, and postprandial inflammation. A total of 1514 papers were analyzed according to their titles and abstracts. From these, 106 articles were chosen for thorough reading. Each selected manuscript was then studied critically and grouped according to its thematic and scientific relevance. Subsequently, 78 were selected, and their references underwent thorough analysis, which resulted in 267 papers chosen for the review. We excluded from the review articles concerning the impact of single food products or single substances on microbiota or inflammation.

## 3. Microbiota

A diverse community of microorganisms inhabits the lumen of the human gut. It is referred to as “gut flora” or “microbiota” and comprises more than 250 species of bacteria, fungi, viruses, and archaea [63,64]. The microbiota of an adult intestine harbors approximately 10^13^ bacterial cells [63,65] and is a dynamic system that changes throughout human life. Moreover, it is highly variable among individuals, and the abundance of specific bacteria species varies depending on the genetics and the structure of the host’s intestinal wall, age, diet, drugs, including antibiotics, and other environmental factors [66,67,68]. The host/gut microbiota relationship is highly mutualistic as the latter plays a crucial role in numerous processes [63,64,69,70].

Since humans produce a very limited variety of enzymes necessary to digest common polysaccharides, the microbiota participates in recovering energy from food, securing additional enzyme activity [71]. Moreover, the microbiota provides hosts with vitamins like thiamine, folate, biotin, riboflavin, and pantothenic acid (abundant in food but also synthesized by gut bacteria). It has been suggested that up to 50% of the daily vitamin K requirement is secured by the microbiota [72,73]. Furthermore, gut flora exerts a protective action against exogenous pathogens, it contributes to maintaining the integrity of the intestinal epithelial barrier [74,75], and is crucial for the development of functional maturation of the gut immune system [76]. The microbiota also influences distant organs outside the intestinal tract. Interaction between the intestinal microbiome and brain is called the gut–brain axis. For instance, it is involved in satiety regulation and hormonal regulation and impacts mood and behavior [57,77,78]. Metabolism of xenobiotics is another aspect of the microbiota’s complex function [79]. Despite extensive knowledge about the physiological role of microbiota, the dysbiosis field has still more questions than answers, and there are just a few authorized health claims regarding restoring the physiological eubiotic balance. These include, for instance, fecal microbiota transplantation in recurrent *Clostridium* difficile infections or inflammatory bowel diseases [80,81].

### 3.1. Healthy Microbiota Composition and Dysbiosis

An essential aspect of microbiota research, which remains a significant challenge to overcome, is the definition of eubiosis. Eubiosis or “healthy microbiota” can be considered the balance of the intestinal microbial ecosystem with a preponderance of potentially beneficial bacteria species. The term is usually used in opposition to dysbiosis [82,83]. Unfortunately, despite rapidly increasing knowledge on microbiota, the definition of eubiosis is still waiting for clarification [83,84].

Human microbiota consist mainly of five phyla of bacteria: the *Firmicutes* [60 to 80%, classes: *Clostridia*, *Bacilli*, and *Negativicutes* (including Gram-negative genres)], the *Bacteroidetes* [20 to 40%, classes: *Bacteroidia*, *Flavobacteria*, *Sphingobacteria*, and *Cytophagia*; with only Gram-negative genres], the *Proteobacteria*, the *Actinobacteria* and the *Verrucomicrobia*, and one *Archaea phyla*, the *Euryarchaeota* [64,85,86,87,88,89,90]. Typically, restricted anaerobes (such as *Bacteroides*, *Clostridium*, *Eubacterium*, *Ruminococcus*, *Peptococcus*, *Fusobacterium*, and *Bifidobacterium*) prevail over facultative anaerobic genera (such as *Lactobacillus*, *Escherichia*, *Enterobacter*, *Enterococcus*, *Proteus*, and *Klebsiella*) [91,92], with *Cyanobacteria*, *Fusobacteria*, and *Spirochaeataceae* [85,93] being less predominant.

Despite being highly variable between individuals, gut microbiota can be divided into three main enterotypes characterized by a dominant bacteria genus: *Bacteroides*, *Prevotella*, or *Ruminococcus* [94]. However, a thorough description of this topic is beyond the scope of this publication. Although the gut microbiota is highly variable, alterations in the composition can lead to the imbalance of gut microbial communities’ activity and disrupt the complex host–microbiota relationship. The alteration of predominant microbiota is called dysbiosis and is associated with the development of numerous diseases [95,96]. For example, fecal microbiota transfer from studies of germ-free animal models established a causal link between dysbiosis and obesity and enteropathies [97,98].

### 3.2. Anti-Inflammatory and Proinflammatory Microbiota

Some gut bacteria have anti-inflammatory or proinflammatory properties. *Akkermansiamuciniphila*, a mucin-degrading strain belonging to the *Verrucomicrobia* phylum, is one of the most frequently described in the literature [63]. It inhabits the intestinal mucus layer and improves intestinal barrier integrity by enhancing mucin production [99] and complex interactions with other bacteria [42,100]. *A. muciniphila* comprises about 3–5% of the microbiota of healthy individuals [32,101] and is strongly correlated with leanness, insulin sensitivity, and reduced low-grade inflammation in human and animal models [100,102,103,104,105,106]. Moreover, *A. muciniphila*-derived extracellular vesicles reduce gut permeability by increasing the expression of tight junctions (TJ) proteins, like occludin (OCLN), in vivo in a murine model and in vitro cultured human epithelial Caco-2 cells [42]. Other examples of bacteria strains with beneficial effects on the intestinal barrier are *Bacteroidesvulgatus* and *Bacteroidesdorei*, which increase TJ expression and produce bacteriocins, proteins that inhibit the growth of specific bacteria, limiting the growth of harmful strains and helping to re-establish healthy microbiota [107,108]. Among commensal bacteria, some strains generating short-chain fatty acids (SCFA) are also considered to have anti-inflammatory properties. SCFA, in addition to being the primary source of energy for the colonic epithelium, also has regulatory functions in energetic metabolism and exert an immunomodulatory effect, maintaining the anti/proinflammatory balance [64,109]. For instance, butyrate, one of the most abundant SCFA in the gut, is anti-inflammatory by stimulating the nuclear transcription factor, peroxisome proliferator-activated receptor gamma (PPAR-γ), which leads to nuclear factor kappa-light-chain-enhancer of activated B cells (NF-κB) pathway inhibition [110,111]. Microbiota species related to increased SCFA production are *Akkermansia*, *Lachnospira*, *Lactobacillus*, *Bifidobacterium*, *Roseburia*, *Ruminococcus*, *Clostridium*, *Faecalibacterium*, and *Dorea* [112]. Among them, *Bifidobacteria* are also reported to maintain intestinal microvilli integrity, promote anti-inflammatory cytokine production, cause maturation of immune cells, stimulate IgA secretion, and have antioxidant properties. However, they do not produce enough endotoxins to enhance proinflammatory cytokine production stimulation [113,114,115,116]. Another SCFA producing strain, *Faecalibacterium prausnitzii*, favors the proliferation of colon epithelial cells and promotes the synthesis of TJ proteins [117]. Bacteria strains that are considered proinflammatory produce more endotoxins. An increase in the relative abundance of Gram-negative lipopolysaccharide (LPS) producing *Proteobacteria*, like *Escherichiacoli*, is observed in dysbiosis [118,119].

### 3.3. High-Fat Diet-Driven Dysbiosis

There is considerable evidence linking dietary fats, colonic microbiota composition, and inflammation. Obesity-related dysbiosis directly links to the high-fat diet (HFD) and manifests in a reduced overall microbiota count, a shift in bacteria species abundance, and an overall increase in gut permeability [120,121]. In a study by Bäckhed et al., gut microbiota transfer from HFD-induced obese mice into germ-free mice caused metabolic syndrome with epithelial barrier dysfunction independent of the recipient diet [122]. Furthermore, several studies on murine and human models have identified the link between obesity, and HFD, with increased endotoxemia, which disrupted the epithelial barrier and enhanced permeation of luminal LPS [33,37,123,124,125]. Moreover, the transplantation of gut microbiota from HFD mice to germ-free mice resulted in increased stimulation of the inflammatory pathway (Nfkb1), which supports the general notion that the presence of diet-induced dysbiosis alone is sufficient to cause inflammation [34]. Finally, HFD in animal models promotes the development of microbiota profiles similar to that of obese male subjects [55,57,126]. Moreover, the study performed by Hildebrand et al. using a murine model prone to diet-induced obesity indicated a shift in the gut microbiota composition in mice fed an HFD independent of obesity development [127].

Changes in gut bacteria species are usually described using the *Firmicutes*/*Bacteroidetes* ratio as a marker of microbiome dynamics [128]. The HFD-driven microbiota composition changes in animal and human models primarily include an increase in the *Firmicutes*/*Bacteroidetes* ratio. Velsquez showed that *Firmicutes* and *Bacteroidetes* were the most abundant phyla in HFD-fed mice and mice fed on a low-fat diet (LFD), comprising 61% and 32% of gut microbiota in LFD mice and 73% and 21% in old-HFD mice, respectively. However, HFD mice had a higher *Firmicutes*/*Bacteroidetes* ratio than LFD mice [129]. The observed changes in *Firmicutes*/*Bacteroidetes* ratio are reported to be driven by increases in *Erysipelotrichales*, *Bacilli*, and *Clostridiales* abundance (all belonging to *Firmicutes* phyla) [130]. Interestingly, Jiao et al. showed that *Clostridia* was the only class that significantly increased in obese rodents [131]. Other gut microbiota changes contributing to the rise of *Firmicutes*/*Bacteroidetes* ratio were increases in *Dorea* and *Ruminococcus* (belonging to *Firmicutes* phylum) abundance [130,131]. Additionally, Velasquez documented that prolonged HFD feeding significantly increased the quantity of *Actinobacteria* and decreased the number of Tenericutes compared to LFD mice (*p* < 0.05) [129]. However, this meta-analysis performed by Jiao et al. indicated that the relative abundance of *Actinobacteria* was reduced [131]. HFD consumption has also been reported to increase the *Proteobacteria* abundance, which are Gram-negative LPS-carrying proinflammatory bacteria [132] and include an *Enterobacteriales* order [133]. HFD is also associated with decreases in *Prevotellaceae* and *Rikenellaceae*, which belong to the *Bacteroidetes phylum* [57,134]. Furthermore, HFD-driven dysbiosis is often related to a reduction in *Bifidobacterium* spp. (the *Actinobacteria* phylum), which is negatively correlated with gut barrier function [133,135].

Interestingly, a thorough literature review has led to inconsistent conclusions regarding microbiota changes driven by HFD (Table 2). According to Fuke et al., this inconsistency might be due to different fat types used in various studies [136]. Similarly, Candido et al. indicated that microbiota changes both in human and murine models are due to the quantity and quality of fat ingested [137]. In mice, dysbiosis typically related to HFD is caused by palm oil supply and not olive or flax/fish oil-based diets [138]. In humans, changes in the intestinal microbiota depend on the type of fatty acids ingested. Omega-3 PUFA intake was directly associated with an increase in *Lactobacillus* abundance, while monounsaturated fatty acid (MUFA) and omega-6 PUFA were inversely related to increased *Bifidobacterium* [139]. Interestingly, according to Wang et al., inconsistent results concerning the association between HFD and microbiota may contribute to the different fibre content in applied diets [53]. However, this requires further investigation.

In observational studies, a significant association between consumed fat and the intestinal microbiome was observed [140]. However, Wolters et al. in a systematic review of interventional studies assessing the impact of changes in dietary fat intake on gut microbiota composition and cardiovascular risk failed to document a strong relationship [140]. The authors highlighted that half of the interventional studies had a relatively short duration which could be a reason why no strong correlation was detected [141,142,143].

Moreover, it should be noted that microbiota research is a constantly and rapidly developing field. However, it still struggles with technical obstacles, including difficulties in collecting and storing samples and analytical weakness of commonly used methods. Wolters et al. and Scarmozzino et al. pointed out that qPCR and FISH methods do not allow a complete taxonomic assessment of the bacteria species inhabiting the gut. More precise next-generation sequencing methods allow better analysis of the composition of gut microbiota. Their application might be necessary to fully elucidate the impact of fat quantity and quality on the intestinal microbiome [83,140].

### 3.4. Disruption of Spatial Microbiota Distribution

The disruption of gut microbiota’s spatial distribution is also worthy of mention. Tomas et al., in a murine model, showed that a 30-day long period of HFD feeding led to the spatial redistribution of microbiota and colonization of the intervillous zone of the ileum (usually described as germ-free) by a dense microbiota, accompanied by drastic changes in the colon microbiota composition [39]. The observed changes might be due to decreased antimicrobial peptide expression, mainly in the ileum. However, the same group has shown that stimulation of PPAR-γ led to the restoration of gut flora’s spatial distribution, whereas PPAR-γ deficient mice presented with ileum colonization [39].

### 3.5. Increased Gut Permeability

HFD-driven dysbiosis is associated with many molecular abnormalities, among which disrupted gut barrier function is the most relevant. Similar to the airways and the genitourinary tract, the gastrointestinal tract is covered with mucosa, which creates a semipermeable barrier allowing the absorption of nutrients and limits the passage of potentially harmful antigens and microorganisms from the gut lumen. However, from the luminal to the basolateral surface, the whole intestinal barrier is very complex, comprising gut microbiota, mucus layer, epithelial cell monolayer, immune cells in the lamina propria, and submucosa [144,145].

Several studies have linked HFD-driven alterations in gut microbiota with increased gut barrier permeability, referred to as the leaky gut syndrome. It is suggested to be caused by reduction in gut barrier–promoting microbes, such as *Akkermansia muciniphila*, *Bifidobacterium* spp., *Bacteroidetes* spp., *Lactobacillus* spp. and *Clostridiales* spp., accompanied by an increase in microbes disrupting gut barrier integrity, such as *Oscillibacter* spp. and *Desulfovibrio* spp. [32,35,146,147,148,149,150,151,152]. Increased gut permeability might also be caused or exacerbated by stimulation of toll-like receptor 4 (TLR4) by LPS, which is discussed later in this article [153,154].

The study by Tomas et al. mentioned above indicated that HFD has a similar impact on the ileum tocystic fibrosis, which decreases the expression of cystic fibrosis transmembrane conductance regulator (Cftr) and the Na-K-2Cl co-transporter 1 (Nkcc1) gene and protein [39]. This reduces ileal chloride secretion, likely responsible for the massive alteration in mucus phenotype, causing increased gut permeability. Tomas et al. stated that both pathologies are associated: PPAR-γ agonists can reverse described changes in Cftr expression and microbiota spatial distribution disruption mentioned previously [39]. Besides, alterations in the mucus layer could be associated with changes in the abundance of *Actinobacteria* phylum, known as mucin-degrading bacteria [155]. This hypothesis is supported by Kim et al., who documented that *Actinobacteria* was inversely related with TJ proteins expression and positively associated with proinflammatory cytokines secretion, suggesting a role in the HFD-induced impairment of the gut barrier [31]. However, this concept should be considered with caution as data on changes in *Actinobacteria* abundance in HFD-fed models are inconsistent.

TJ play a crucial role in the intestinal barrier function along with the mucous layer. They are a network of transmembrane protein strands that link laterally adjacent cells near the epithelium apical surface [156] and include, among others, claudins, OCLN, cingulin, tight junction proteins 1 and 2 (TJP1 and 2), TJ-associated MARVEL domain-containing proteins (TAMPs), and junctional adhesion molecules (JAMs) [157]. Some gut microbiota genres are proven to have a stimulative effect on TJ proteins expression. For example, *Bifidobacterium* spp. and *Lactobacillus* spp. (both often decreased in HFD) are related to the stimulation of the TJ proteins enterocyte gene expression like cingulin, OCLN, TJP1, and TJP2 [158,159,160,161]. Furthermore, *Akkermansia muciniphila* induces Tjp1 and Ocln gene expression in addition to counteracting HFD-induced thinning of the intestinal mucous layer [32,35]. In contrast, the HFD-increased abundance of *Oscillibacter* spp. directly correlates with depressed expression of TJP1 in the gut epithelium [162].

Another factor involved in maintaining gut barrier integrity is IL-17-producing T helper 17 cells (Th17) [50]. The loss of Th17 cells is observed both in metabolic disorders and HFD. According to Garidou et al., it is caused by an impairment in antigen-presenting cell function in the lamina propria of the small intestine following HFD-induced dysbiosis [38]. Th17 response failure contributes to increased gut permeability and favors translocation of LPS into the circulatory system [38,163].

### 3.6. Decreased SCFA

Gut microbiota obtains energy mainly by fermentation of non-digested carbohydrates like fiber, generating SCFA [164]. Acetate, propionate, and butyrate comprise 95% of SCFA in the colon and feces in humans [165]. SCFA have multiple physiological functions. They are an essential energy source for the colonic epithelium and liver gluconeogenesis and play a role in regulating energy metabolism and immune system modulation [109]. The latter is obtained mainly by stimulation of free fatty acid receptors. Four types of SCFA receptors (FFAR—free fatty acid receptors) have been identified, GPR43/FFAR2, GPR41/FFAR3, GPR109A, and Olfr7. SCFA receptors are present in multiple tissues, including the colon, small bowel, adipose tissue, liver, skeletal muscle, and pancreatic beta-cells [166], however, a thorough description of this topic is beyond the scope of this paper.

Butyrate reduces local inflammation in the gut and intestinal permeability through multiple mechanisms, including stimulation of mucin synthesis, increased TJ expression, and inhibition of the NF-κB pathway via stimulation of PPAR-γ [110,167,168,169,170,171,172,173]. Furthermore, several studies demonstrated that HFD reduces overall SCFA more than LFD [29,41,60]. Moreover, Agus et al. indicated that high-fat and high-sugar diets (e.g., WD) reduce the expression of the GPR43 in murine models compared to controls [29].

### 3.7. Endocannabinoid System

Another link between dietary fat, gut microbiota, and intestinal permeability is the endocannabinoid system (EC) of the gut [174]. The EC is a known contributor to the hedonic regulation of food intake in mammals [175]. It has been demonstrated that EC is also involved in the regulation of glucose and energy metabolism, and specific gut microbiota species can modify EC activity. Furthermore, obesity-related and HFD-related dysbiosis is characterized by increased EC activity [176], followed by increased gut permeability and LPS translocation [33,177]. Accordingly, Muccioli et al. documented that blockage of the cannabinoid receptor CB1 lowered gut permeability in obese mice, whereas CB1 stimulation increased permeability markers in vivo and in vitro [177].

### 3.8. Endotoxemia

The LPS is a proinflammatory molecule produced by some Gram-negative bacteria. It comprises a part of the Gram-negative bacterial outer membrane and has been identified as a key contributing factor in the initiation and progression of low-grade inflammation [33]. As described above, HFD-induced dysbiosis is characterized by increased gut barrier permeability, known as leaky gut, enhancing LPS translocation from the gut lumen into the bloodstream. However, this is not the only mechanism underlying increased immune system stimulation with LPS, as increased gut LPS production is observed in HFD-associated microbiota alterations along with leaky gut.

In 2007, Cani et al. indicated that HFD significantly increased LPS-containing bacteria abundance in the gut [33]. It has also been shown that the number of *Enterobacteriaceae* was elevated in the feces of HFD mice. Furthermore, in the same study, LPS production was increased when the fecal microflora from wild type mice was cultured in high-fat-containing media rather than in low-fat-containing media [47]. In agreement with these findings, Jeong et al. showed that HFD increased both plasma and fecal endotoxin levels and induced the growth of *Enterobacteriaceae* and endotoxin production in vitro [45]. Moreover, Crawford et al. demonstrated an increase in LPS concentration in HFD rat plasma [30].

Interestingly, Kim et al. showed that *Lactobacillus plantarum* LC27 and *Bifidobacterium longum* LC67 supplementation reduced HFD-driven *Firmicutes* and *Proteobacteria* populations in gut microbiota and fecal LPS production [48]. Based on this, it can be assumed that HFD may induce LPS production by providing a favorable condition for the proliferation of Gram-negative bacteria in the intestine, while probiotics can reverse this effect.

Increased gut permeability and enhanced gut LPS production associated with an HFD results in higher plasma endotoxin concentration, known as “metabolic endotoxemia” [130]. LPS on its own has multiple adverse effects on gut function, as it promotes intestinal inflammation, disrupts TJ organization via specific signaling pathways, directly causes intestinal epithelial cells shedding without compensatory TJ-resealing, and may induce OxS in intestinal epithelial cells, mitophagy, and mitochondrial failure [153,178,179]. LPS indirect action is mainly mediated by a TLR4-CD14–dependent proinflammatory response [180]. In the study by Park et al., increased penetration of FITC-dextran (a marker of increased gut permeability) through the gut barrier and lowered expression of intestinal TJ in HFD mice were associated with a significantly higher serum concentration of LPS receptor (CD14) and colonic TLR4 mRNA expression compared to the control group [56].

TLR4 belongs to the family of pattern recognition receptors, and its activation leads to proinflammatory cytokine release and increased gut permeability [181]. TLR4 is present in various immune cells (monocytes, macrophages, and Kupffer cells) and multiple other cells such as endothelial cells, adipocytes, and hepatocytes [182]. The recognition of LPS by TLR4 is mediated via the LPS-binding protein (LBP). LBP is the membrane protein cluster of differentiation 14 (CD14) co-receptor [182]. An interaction of LPS with LBP allows the activation of TLR4, which triggers a signaling cascade that results in the activation of a focal adhesion kinase (FAK) in intestinal epithelium cells. Subsequently, FAK enhances the activation of the of myeloid differentiation primary response gene 88 (MyD88) and kinase 4 related to the interleukin-1 receptor (IRAK4), increasing intestinal permeability [153,154] and activation of downstream signaling pathways, including NF-kB and mitogen-activated protein kinase (MAPK), promoting inflammation [183,184,185]. As a result of NF-kB pathway activation, increased gene expression of tumor necrosis factor-α (TNF-α), IL-6, inducible NO synthase (iNOS), and monocyte chemotactic protein-1 (MCP1) occurs [186].

Cani et al. challenged LPS receptor knockout mice (CD14 knockout mice-CD14KO) with an HFD or with a chronic infusion of low doses of LPS, or both, to demonstrate the causative link between the CD14-TLR4 pathway and inflammation. They revealed that CD14KO mice were resistant to HFD or LPS-induced development of the inflammation [33]. These results agree with the findings by Kim et al., who demonstrated that HFD did not affect the proinflammatory cytokine levels in the TLR4-deficient murine model [47].

Since increased LPS plasma concentration via TLR4 favors the release of TNF-α and interleukins IL-1 and IL-6, it is conceivable that the alterations in the gut microbiota caused by HFD may play a pivotal role in the induction of low-grade inflammation [47,135]. This general notion is supported by the fact that antibiotic treatment in mice models with diet-induced obesity reduced gene expression of inflammatory markers and concentration of lipid peroxides (a marker of OxS) in the visceral adipose tissue, which reproduced the effect of knockout of CD14 receptors [33,37].

HFD disrupts gut barrier function and microbiota composition, which results in endotoxemia and leads to bacteremia, both are referred to as “metabolic”. The presence of gut-derived bacteria in blood and white adipose tissue has been reported after only one week of HFD. Furthermore, it was prevented in mice lacking the microbial pattern recognition receptors Nod1 or CD14, suggesting common mechanisms of both metabolic endotoxemia and metabolic bacteriemia [40].

### 3.9. Bile Acids

An interesting yet complicated aspect of HFD-associated inflammation related to gut dysbiosis is the alteration in bile acids (BAs) secretion and metabolism. Firstly, it is proposed that an HFD-driven increase in the release of BAs may have proinflammatory effects on the microbiota composition, as BAs have been identified as factors changing caecal microbiota composition in male rats [187]. This shift might result from the BAs ability to promote BAs-metabolizing bacteria development, preventing the growth of bacteria sensitive to BAs [188]. For example, HFD increases the abundance of *Desulfovibrionales*, specifically *Bilophila wadsworthia*, the growth of which is stimulated directly by colonic levels of taurocholic acid [151,189,190,191]. *Bilophila wadsworthia* produces hydrogen sulfide which inhibits butyrate oxidation, disrupting the energy balance in enterocytes leading to their damage and causing intestinal epithelial cells hypoplasia and hyperpermeability, finally resulting in leaky gut and inflammation [151]. Moreover, it has been shown that feeding rats with cholic acids at levels similar to those observed while on HFD significantly altered the microbiota at the phylum level and resulted in an increased abundance of *Firmicutes* and a reduced abundance of *Bacteroidetes* [187]. Thus, BAs play an essential role in the regulation of microbiota composition.

A study using antibiotics was performed to confirm the causal link between gut microbiota, BAs, and inflammatory status in HFD models. Antibiotic treatment altered gut microbiota composition in HFD-fed mice and reduced the deoxycholic acid (DCA) and taurodeoxycholic acid (TDCA) levels, along with proinflammatory cytokine levels [130].

### 3.10. Inflammation-Related Gene Expression

HFD-driven microbiota alterations in various mechanisms disrupt the expression of inflammation- and metabolism-related genes. NF-κB pathway-related stimulation of TNF-α, IL-1β, IL-6 production is most commonly mentioned. HFD limits AMP-activated protein kinase (AMPK) activation, which improves lipid metabolism and insulin sensitivity, whereas probiotics supplementation (*Lactobacillus plantarum* LC27 and *Bifidobacterium longum* LC67) increased HFD-suppressed AMPK activation, favoring proper lipid biotransformations. Furthermore, it has been shown that HFD induced iNOSs and cyclooxygenase 2 (COX-2) expression, while treatment with LC27 and LC67 alleviates this effect [48].

### 3.11. Neutrophils and Macrophage Activation

HFD-driven dysbiosis is directly associated with the activation of immune cells, among which neutrophils and macrophages have been most thoroughly examined. Brandsma et al. reported that the transplantation of proinflammatory microbiota to antibiotic-treated LDLR−/− mice (low-density lipoprotein receptor knockout mice) accompanied by HFD resulted in increased numbers of circulating monocytes and neutrophils in plasma compared to the control group (HFD-fed LDLR−/− mice transplanted with regular LDLR−/− mice microbiota) [41].

The increase inboth *Firmicutes* and *Actinobacteria* on HFD is positively correlated with the gene expression of proinflammatory cytokines in colonic macrophages (TNF-α, IL-1β, and IL-6) [31]. It is thus suggested that the observed changes are due to TLR4 activation as described above [192,193]. TLR-bearing macrophages have also been identified in adipose tissue, skeletal muscles, and the liver. Moreover, adipose and muscle cells can express TLR4. Their stimulation increases the production of proinflammatory adipokines and myokines, respectively [62,194,195]. Interestingly, Guo et al. reported a persistent increase in liver macrophages four weeks after the withdrawal of injections with subclinical doses of LPS and suggested that exposure to super-low doses of LPS may contribute to the establishment of a sustained “memory” inflammatory state [62].

Similar to macrophages, LPS treatment leads to a significant elevation of neutrophils in mice compared to controls. Guo et al.showed that the injection of super-low-dose LPS resulted in an elevation of liver neutrophils and neutrophil myeloperoxidase (MPO) along with increased levels of neutrophil chemoattractants and MAPK, the latter two possibly contributing to neutrophil infiltration. In neutrophils, persistent memory effects of subclinical low dose endotoxemia were documented, as sustained neutrophil infiltration, elevated liver MPO, and chemoattractants concentration. MAPK activation was observed even after one month after the stoppage of LPS administration. Moreover, this was accompanied by a sustained apoptotic response in the liver, as MPO and MAPK play a crucial role in cellular apoptosis regulation [62].

### 3.12. Decreased Production of Antimicrobial Peptides by Paneth Cells

Finally, Guo et al. indicated that HFD altered gut microbiota composition after only eight weeks, followed by decreased Paneth cell antimicrobial peptides, such as lysozyme and Reg IIIγ. An increase incirculating inflammatory cytokines, like interferon-γ (IFN-γ) and TNF-α, was observed during HFD feeding for 16 weeks [43]. These findings emphasize the potential role of antimicrobial peptides in the development of a proinflammatory milieu in the gut.

## 4. Non-Dysbiosis Related Inflammation

As described above, HFD leads to alterations in gut microbiota composition, resulting in an enhanced proinflammatory state. However, it has been shown that fat per se can stimulate local and systemic inflammation, contributing to inflammation termed as ‘metabolic’ [58,193]. Napier et al. showed that germ-free mice had higher sepsis severity and mortality when on a WD than when fed a standard chow diet. They stated that microbiota is not required for the enhanced pathologies related to WD and assumed that this result may be driven by dietary constituents like fatty acids [58].

Postprandial inflammation is worth mentioning as modern humans spend more than 16 h per day in the fed state. According to the definition, the postprandial state, also referred to as the fed state, occurs after food ingestion, embodies the digestion and absorption of nutrients, and is considered to last 6–12 h [196]. The postprandial state is correlated with several chronic systemic low-grade inflammation diseases such as type 2 diabetes mellitus, atherosclerosis, or non-alcoholic fatty liver disease [197,198,199]. As the immune system can react to the acute elevation of nutrients, including carbohydrates and fatty acids, the evidence for direct food-induced inflammation is most compelling in the postprandial state [193].

### 4.1. Saturated Fatty Acids

Some nutrients are more strongly related to a proinflammatory response than others, for example, with saturated fatty acids (SFA), the quantity and quality consumed are considered to be a major determinant of the magnitude of postprandial inflammation [193]. Shi et al. and Ohashi et al. showed that SFA can exert a similar molecular effect to LPS and activate TLR4, leading to the release of proinflammatory cytokines, gut barrier function impairment, and disruption of cellular metabolism [200,201]. Interestingly, PUFAs, especially the n-3 to n-6 ratio, have an impact on the meal-related inflammatory response, with n-3 PUFA shown to suppress postprandial inflammation, while n-6 PUFA contributes to inflammation [202,203,204,205,206,207].

### 4.2. Oxidative Stress

One of the possible mechanisms by which HFD can exert its proinflammatory effect is the stimulation of OxS. Supporting this general notion, Gulhane et al. reported that HFD induced the expression of genes that are considered markers of endoplasmic reticulum (ER) stress (unfolded protein response (UPR) signaling molecule sXbp1, ER chaperone Grp78, and ERAD chaperone Edem1), which is closely related to OxS and more thoroughly described below [54]. Moreover, few aspects of HFD-driven OxS should be considered. On the systemic level, fat overconsumption triggers mitochondrial β-oxidation of free fatty acids, subsequently increasing ROS production, which can cause a proinflammatory response [208,209,210]. This is mediated, for instance, by activating NF-κB causing over expression of proinflammatory cytokines like IFN-γ, TNF-α, and iNOS [211]. Due to the increased expression of the latter, overproduction of nitric oxide, followed by the accumulation of reactive nitrogen species (RNS), occurs in addition to ROS production [212]. However, at the gut level, the oxidation of PUFAs containing double bonds occurs, these oxidation products diffuse freely across the apical membrane of enterocytes and induce intracellular OxS. In addition, it is assumed that oxidized PUFA derivatives peroxidize cellular membrane phospholipid components directly from the luminal compartment. Both mechanisms result in OxS and contribute to a disruption of the gut barrier, not only by the induction of proinflammatory pathways in the mucosa but also by changes in the enterocyte shedding–proliferation axis and alterations in the TJ expression [213,214]. In agreement, the findings by Li et al. showed how palmitic acid or palmitic acid combined with TNF markedly increased ROS production and induced the myosin light chain kinase (MLCK) TJ regulatory pathway in vitro in HCT116 cell culture [49]. Interestingly, these effects were markedly reduced in the presence of an ROS scavenger. Moreover, Park et al. indicated that the 8-hydroxy-2′-deoxyguanosine level (a marker of OCS) was higher in the HFD group compared to the control. In contrast, blood total antioxidant capacity was lower in the HFD group, highlighting another potential mechanism by which HFD is implied in OCS [56]. Lastly, the loss of XIAP (X-linked inhibitor of apoptosis protein) function is observed on HFD, which contributes to increased inflammasome activity, apoptosis, and increased OxS, accompanied by suppression in Nrf2 (nuclear factor erythroid-derived 2-like 2)-mediated anti-oxidative activity [215,216,217,218,219].

### 4.3. Endoplasmic Reticulum Stress

ER stress is closely associated with OxS. This term refers to impairment in ER function, which triggers the UPR, a tightly organized collection of intracellular signal transduction reactions aimed at restoring protein homeostasis and alleviating the accumulation of misfolded proteins in the ER. The UPR activation implies an increased expression of ER chaperones, inhibition of protein entry into the ER by arresting mRNA translation, and stimulates retrograde transport of misfolded proteins from the ER into the cytosol for ubiquitination and lysosomal degradation [220]. The literature provides multiple links between ER stress and the emergence of inflammatory responses. Among others, ER stress is shown to activate proinflammatory pathways involving IκB kinase (IKKβ) and c-Jun NH2-terminal kinase (JNK), transcriptional factor CREB-H, and induction of ROS production. In turn, the inflammatory response, dysregulation of adipokine secretion, adipose tissue expansion, and SFA can enhance ER stress contributing to self-propelled mechanisms of inflammation [221,222,223,224,225,226,227]. Gulhane et al.hypothesized that HFD-associated ER stress occurs in gut secretory goblet cells, causing an inflammatory response and reducing the synthesis and secretion of mucous proteins. This hypothesis is supported by the fact that in vitro in intestinal cells, long-chain SFA directly increased ER stress leading to protein misfolding, impairment in goblet cell differentiation, and Muc2 expression, accompanied by a loss in TJ proteins [54].

### 4.4. TLR4 and NF-κB Pathway

TLR4 can also be stimulated directly by SFA, showing a common mechanism of action of dysbiosis and fat alone [137]. It has been proposed that this may be due to structural similarity between dietary SFA and the lipid A component of LPS [228]. SFA, but not PUFAs, were shown to stimulate the NF-κB-pathway in a TLR4-dependent manner. Subsequently, it has been attributed to TLR4-homodimerization via lipid raft incorporation [229,230]. The investigation of various SFA’ proinflammatory potency revealed that it varies depending on chain length, with lauric acid showing the most significant proinflammatory activity. In contrast, myristic acid and stearic acid appeared to have surprisingly little proinflammatory impact. In comparison to SFA, MUFAs and PUFAs failed to activate TLR4 signaling. Moreover, the TLR4-dependent ability of PUFAs to block inflammatory responses induced by LPS or lauric acid has been demonstrated [231,232]. HFD rich in SFA not only directly activates TLR4 but also induces TLR4-mRNA and protein expression in intestinal tissues. Interestingly, TLR4/NF-κB activation gradually increases with the number of HFD administration days. Furthermore, according to Wang et al., HFD induced TLR4 in a shorter period than needed for bacterial enhancement and LPS release, thus avoiding the interference of LPS-driven TLR4 stimulation [233].

According to Hu and Zhang, HFD downregulates autophagy markers levels (Atg5, Atg12, and LC3B) while increasing p62 accumulation [44]. The latter is associated with reduced fusion of autophagosome and lysosome [234,235]. AlthoughTLR4 knockout does not affect Atg5, Atg12, LC3B, and p62 expression, it reconciles HFD-driven changes in autophagy [44].

Interestingly Hu and Zhang also reported that in the murine TLR4-knockout model, phosphorylation of IKKβ, JNK, and mTOR induced by HFD is alleviated, and production of ROS is reduced [44], indicating that multiple mechanisms disrupted by HFD, like ER metabolism or TLR4 pathway, lead to common proinflammatory effects.

### 4.5. TNF-α and IL-6

Similarly, as for dysbiosis, the TNF4-NFκB pathway stimulated by HFD alone increases TNF-α and IL-6 expression. It was reported that serum TNF-α and IL-6 elevation occurs in a postprandial state because of high-fat meals. Moreover, it is associated with hepatic production of acute-phase proteins such as C-reactive protein (CRP) and increased myeloperoxidase activity [51,56,236,237,238]. TNF-α and IL-6 due to HFD are increased not only in plasma; Wang et al. also demonstrated their elevation in intestinal tissue, which is partially responsible for local intestinal low-grade inflammation [233]. As expected, in vitro studies on 3T3-L1 adipocytes indicated that increased IL-6 production was caused by certain SFAs (myristic and palmitic acids). Furthermore, the addition of LPS to the culture medium was correlated with a significant accumulation of IL-6 for each studied SFA (myristic, palmitic, linoleic, and linolenic acids) [61]. This supports the concept that fat alone and fat-driven dysbiosis are associated, yet partially independent, factors contributing to low-grade systemic and intestinal inflammation.

### 4.6. Increased Gut Permeability and Decreased Tight Junctions

Tomas et al. reported that HFD stimulates microbiota spatial disruption along with gut mucous layer disruption, the latter being mediated by reduced Cftr expression. Researchers have shown that both phenomena can be reversed by PPAR-γ agonists, as well as switching the diet back to a regular one, which suggests a close relationship between them [39]. However, these findings do not fully determine whether the observed mucus layer impairment was due to microbiota presence in the small intestine or rather by HFD itself, which requires further investigation.

The recent literature review by Rohr et al. stated that dietary fats directly modulate intestinal barrier integrity, thus stimulating gut permeability and contributing to leaky gut [180]. It is said to be related to reduced TJ expression [37,239].

In corroboration, continuous feeding of mice with unsaturated fatty acids significantly reduced TJ expression and increased flux of FITC-dextran. Interestingly such results were not obtained with SFA [240]. Furthermore, Murakami et al. observed reduced TJ protein expression in three different HFD diets (lard-based, soybean oil-based, and mixed) in mice [59]. However, it seems necessary to highlight that most available studies do not consider the impact of microbiota and are focused on the relationship between HFD, obesity, and impairment of gut barrier function. Supporting this notion, Rohr et al. have drawn a conclusion based on Suzuki and Hara’s study that reduced TJ originated with the dietary fat itself, not due to the diet’s metabolic consequences, including obesity [180,241]. For this reason, like the disrupted mucous layer, it is difficult to establish whether aberrations in TJ result from HFD alone or HFD-driven dysbiosis.

Gut permeability through another mechanism is undoubtedly caused by dietary fat. Absorption of fats involves chylomicron formation in the postprandial period. After high-fat meal consumption, an accumulation of chylomicrons in the intestinal mucosa intercellular space can increase the local pressure, resulting in TJ loosening between the enterocytes or even basal membrane perforation [242,243,244]. The compromised gut barrier becomes more permeable for LPS translocation, favoring inflammation [182], which is discussed in detail below.

Finally, fat absorption activated mast cells in the intestinal mucosa of rats, resulting in an increased release of mast cells mediators, including histamine or prostaglandin D2. This process is positively correlated with an elevation of transcellular and paracellular intestinal permeability [245].

### 4.7. Gut Microbiota—An Innocent Passerby

It seems important that HFD is linked to inflammation by favoring the translocation of LPS from the gut into the bloodstream. It is estimated that gut microbiota contains more than 1 g of LPS [246]. In the postprandial period, the compromised intestinal barrier can promote LPS translocation even in the state of a healthy gut microbiome, which makes it an “innocent passerby” of HFD-associated inflammation.

Dietary fat can also enhance the plasma LPS concentration independent of intestinal permeability. LPS consists of an O-antigen, core polysaccharide, and immunogenic lipid A, a branched-chain SFA. For this reason, LPS is incorporated into micelles and takes part in chylomicrons formation in enterocytes via its lipid A tail and is subsequently delivered to the circulatory system [247]. Furthermore, it also stimulates lipid raft-mediated endocytosis [136,247,248].

Cani et al. demonstrated that serum LPS levels varied between fasted and fed mice [33]. A similar response was observed in men consuming high-fat meals, increasing postprandial LPS levels compared to fasted individuals [249]. A higher postprandial plasma LPS concentration has been observed in healthy adults on HFDs rich in saturated fats compared to individuals fed polyunsaturated fats [250]. Interestingly, Mani et al. indicated that circulating concentrations of endotoxin were dependent on the type of fat ingested, as pigs fed coconut oil (rich in SFA) had higher LPS serum levels than those given vegetable oil and fish oil. This was independent of the overall intestinal integrity or permeability, as testing freshly isolated ileal samples showed that the gut barrier was not affected [248]. Seemingly inconsistent findings suggest that HFD-driven metabolic endotoxemia may be caused by the combination of intestinal hyperpermeability and diet-induced LPS translocation.

### 4.8. Bile Acids

Another subsequent effect of dysbiosis and high dietary fat intake is BA alterations. BAs are mostly reabsorbed from the gut lumen, undergoing enterohepatic recirculation, with only 5 to 10% of BAs not reabsorbed and undergo biotransformation to secondary BAs. For this reason, the composition of BAs in the small intestine is similar to the biliary pool. In contrast, the BA profile in the colon differs significantly, as it comprises secondary BAs. The biotransformations mentioned above include hydrolysis of conjugated BAs to free BAs and glycine or taurine by bile salt hydrolase. Cholic acid and chenodeoxycholic acid (CDCA) undergo 7α-dehydroxylation resulting in DCA and lithocholic (LCA) acid formation, respectively, while ursodeoxycholic acid (UDCA) undergoes 7β-dehydroxylation to lithocholic acid [251,252].

Intestinal epithelial cells are usually resistant to BAs’ solubilizing effects under physiologic conditions. It has been shown that HFD increases total BAs and total secondary BAs along with elevated BA concentration in the caecum, while chronically high gut and fecal BA concentrations can reduce gut barrier integrity. Thus, the HFD-associated gut hyperpermeability can be related to increased BA secretion [55,231,232,233]. Moreover, Stenman et al. reported that a 10-fold increase in BA synthesis on HFD was associated with enriched bile composition in hydrophobic BAs such as DCA, LCA, and CDCA, which are indicated to stimulate intestinal permeability when administered at high concentrations [253].

Primary BAs are reported to impair intestinal barrier integrity in Caco-2 cells [241]. In the same cell line, the administration of high DCA and CDCA concentrations led to stimulation of the epidermal growth factor receptor (EGFR) pathway, resulting in TJ disruption and enhanced intestinal permeability, which was reversed by EGFR inhibitor treatment [254]. Moreover, Caco-2 cells incubation with supra-physiologic hydrophobic BA concentrations enhanced ROS generation in enterocytes resulting in OxS and subsequent aberrations in TJ [255,256,257]. Another mechanism related to BAs by which HFD can modulate intestinal integrity is the structural similarity of LCA and the product of linoleic acid oxidation (13-hydroxyoctadecadienoic acid) [258]. The latter is one of the most abundant SFA in WD. Interestingly, in contrast to hydrophobic BAs, hydrophilic BAs such as UDCA enhance the intestinal epithelium integrity in mice through multiple mechanisms, including reducing OxS [259,260,261,262].

Secondary BAs play a role in regulating lipid signaling pathways and immune system activity, partly through BAs receptors, such as Takeda G protein-coupled BAs receptor-1 (TGR5), farnesoid X receptor (FXR), and pregnane X receptor (PXR) [263,264,265,266,267]. For example, macrophages of the gastrointestinal tract can be activated by the binding secondary BAs with the TGR5 receptor [268]. Interestingly, the effect of TGR5 stimulation depends on macrophage phenotype, either proinflammatory M1 or anti-inflammatory M2. TGR5 stimulation induces a partial transformation from the M1 to the M2 phenotype, favoring an anti-inflammatory response, subsequently inhibiting proinflammatory cytokines, such as TNF-α and IL-6 [269], while BAs may exert various effects on the inflammatory status depending on the concentration in the gut lumen. At relatively low concentrations (e.g., <50 μM), secondary BAs may have an anti-inflammatory effect on the colon by decreasing proinflammatory cytokine release [270]. However, at high physiological concentrations (i.e., in HFD), secondary BAs can cause, among others, OxS, DNA damage, inflammation and activation of NF-κB pathway, and apoptosis [265,271,272,273]. Moreover, the detergent property of DCA causes membrane disruption resulting in activation of protein kinase C and a release of arachidonic acid, which are metabolites with strong proinflammatory properties [274,275]. All described processes contribute to the production of proinflammatory cytokines, such as IL-6 and TNF-α, that promote inflammation and cause inactivation of FXR, subsequently leading to the proinflammatory status in the colon [264,265,276,277].

### 4.9. Macrophages and Neutrophils Activation

The recruitment of monocytes/macrophages to inflammatory sites is mediated by the chemokine (C-C motif) ligand 2 (Ccl2), also referred to as MCP1 [278]. According to Kawano et al., Ccl2 and Ccr2 (encoding a chemokine receptor for Ccl2) expression was significantly increased in the colon of mice fed an HFD. Moreover, it has been shown that a deletion of Ccl2 in enterocytes prevents HFD-induced infiltration of proinflammatory macrophages, stimulation of inflammation, and subsequent leaky gut. These data strongly suggest that macrophages enhance gut permeability in a Ccl2/Ccr2-dependent manner [46].

Also, immune cells present in adipose tissue play an important role in stimulating and sustaining HFD-associated low-grade inflammation. HFD is proved to stimulate macrophage infiltration of adipose tissues and induce the expression of proinflammatory cytokines, including TNF-α, IL-1, and IL-6, not only in plasma but also in the adipose tissues [279,280]. Elevated expression of macrophage-related genes in adipose tissues was observed within three weeks of being on an HFD in murine models [281]. Moreover, macrophages present in obese mice adipose tissue express genes associated with the M1 (proinflammatory) phenotype, whereas macrophages from lean mice adipose tissue belong to the M2 (anti-inflammatory) phenotype [282]. Similarly, as in dysbiosis-associated immune cell activation, chronic inflammation in adipose tissues of individuals with obesity is non-resolving and persists over a long period, indicating a memory effect in the macrophage populations [283].

Peripheral blood mononuclear cells (PBMC) have a dynamic inflammatory response to nutrient intake. HFD-related plasma concentration of free fatty acids is known to increase TNF-α and IL-6 mRNA expression in PBMC, including macrophages [284]. Besides, in vitro stimulation of PBMC with palmitate can activate autophagy pathways [285]. Interestingly, Lowry et al. showed that HFD induces macrophages’ autophagy pathway in rabbits, however, autophagy was impaired: the fusion of autophagosomes with lysosomes and the maturation of this complex was disrupted. Thus, HFD leads to modification of the autophagy pathway, which possibly contributes to enhanced proinflammatory monocyte-macrophage polarization [286,287]. Moreover, an HFD impairs the activation of inflammatory and autophagy pathways in macrophages before metabolic and inflammatory changes occur in adipose tissue, highlighting the importance of immune cells in HFD-mediated inflammation in peripheral tissues [286].

Neutrophils and macrophages infiltrate adipose tissue upon commencement of an HFD. For example, Talukdar et al. showed that the neutrophil content in mice adipose tissue rises rapidly after HFD introduction compared to the control group, accompanied by an elevation in the expression of proinflammatory neutrophil elastase. Consistent with the abundance of neutrophil and elastase expression, the latter’s activity was also significantly higher in the HFD group. Moreover, adipose tissue neutrophils secreted chemokines and cytokines favoring macrophage infiltration, which could aggravate the chronic low-grade inflammation. In the same study, HFD induced neutrophil infiltration and increased neutrophil elastase concentration and activity in the mice liver [52]. These findings highlight the importance of cooperation between neutrophils and macrophages in stimulating and sustaining the low-grade inflammation upon HFD.

### 4.10. The Decrease in Gut Peptides

Gut peptides like ghrelin, cholecystokinin, vasoactive intestinal peptide (VIP), glucagon-like peptide 1 (GLP-1), and peptide YY (PYY) are conventionally related to appetite regulation and gut motility and secretion. However, more recent studies have demonstrated their role in mucosal immune tolerance and barrier integrity. Moreover, they have been shown to have anti-inflammatory properties, possibly due to preventing bacterial translocation in the gut by increasing TJ expression, suppressing proinflammatory cytokines released by T cells, monocytes, and dendritic cells, and preventing macrophage activation and migration [192]. HFD, therefore, reduces gut peptide secretion by enteroendocrine cells and downregulate their signaling pathways. In the murine model, HFD impaired ghrelin secretion and sensitivity to exogenous ghrelin compared to lean mice [288]. In rats, HFD was associated with decreased GLP-1 production and reduced GLP-1 receptor sensitivity [289]. These results strongly supported the general notion that changes in gut peptides secretion driven by HFD can contribute to diet-induced low-grade organ-specific and systemic inflammation.

## 5. Summary

In summary, HFD-associated low-grade inflammation is a complex phenomenon (Figure 2). Dysbiosis reduces overall SCFA production in the gut and expression of SCFA receptors, which may enhance local inflammation and intestinal permeability [29,41,60]. Furthermore, HFD-related dysbiosis is associated with increased EC activity [176], contributing to increased gut permeability and LPS translocation [33,177]. Moreover, unfavorable changes in gut flora caused by HFD also result in increased LPS production [33,47,48]. HFD-associated microbiota changes can also disrupt the expression of inflammation- and metabolism-related genes. Interestingly, changes in microbiota observed on HFD are followed by decreased Paneth cells antimicrobial peptides, which can stimulate the development of a pro-inflammatory milieu in the gut [43]. HFD enhances oxidative stress by increasing ROS and RNS production [208,209,210,211,212], stimulating closely related ER stress [220,221,222,223,224,225,226,227], downregulating gut peptide signaling pathways, and reducing their secretion by enteroendocrine cells, thereby contributing to a leaky gut [192,288,289]. Dysbiosis and inflammation independently disrupt the mucus layer and reduce TJ expression, resulting in a leaky gut, increased LPS translocation, and metabolic endotoxemia [30,33,47,48,135,180,240,241,249,250]. Moreover, TLR4 can be activated by LPS but also by SFA. This clearly shows that NF-κB stimulation and subsequent pro-inflammatory cytokine production can be caused both by changes in gut flora and HFD, proving convergence in the action of both mechanisms [47,56,137,153,154,180,229,230,231,232,233]. Neutrophil and macrophage activation is another common aspect of dysbiosis and HFD [28,37,42,48,58,257,258,259,260,261,262,263,264,265]. An interesting accent of HFD and dysbiosis convergence is related to bile acids, the increased secretion of which can impair gut barrier function and have pro-inflammatory effects [59,241,253,254,255,256,257], as well as pro-inflammatory effects on the microbiota composition [151,187,189,190,191].

## 6. Conclusions

HFD-associated low-grade inflammation results from overlapping effects of dysbiosis and high fat intake, with crosstalk between microbiota and fat-driven inflammatory processes independent of gut flora involving multiple mechanisms. For this reason, it is difficult to determine what occurs first, thus, HFD-driven dysbiosis and HFD-related inflammation should be considered partially independent but closely correlated mechanisms.

## Figures and Tables

**Figure 1 cells-10-03164-f001:**
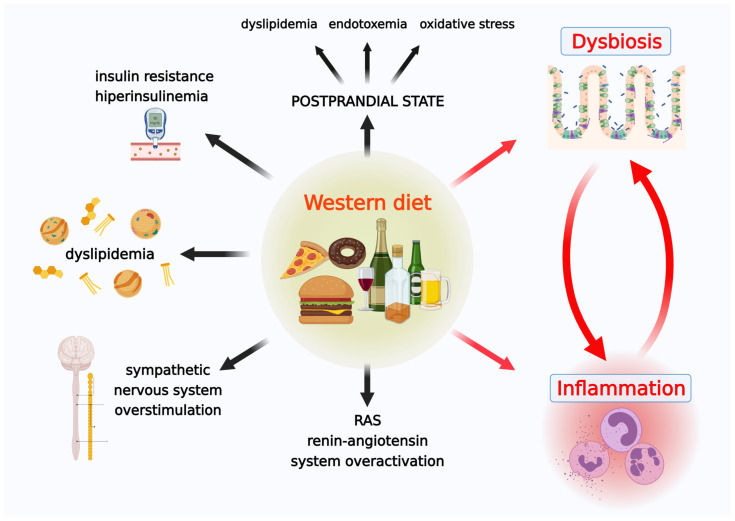
Western diet-associated pathologies.

**Figure 2 cells-10-03164-f002:**
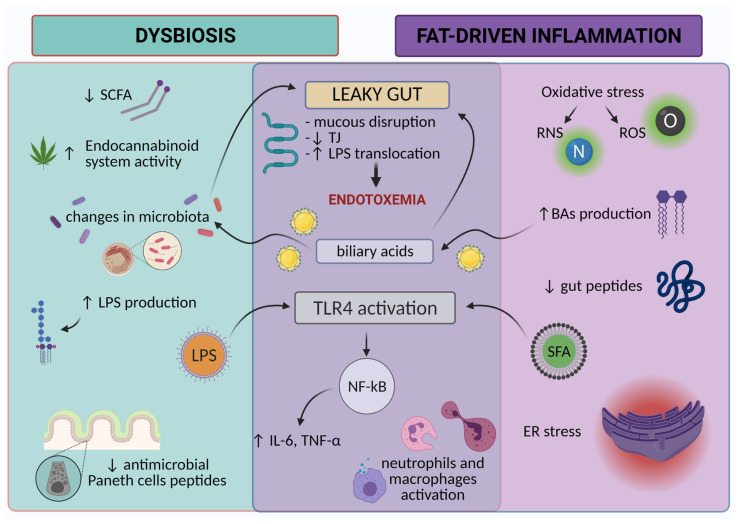
Western diet-associated pathologies. SCFA, short-chain fatty acids; LPS, lipopolysaccharide; TJ, tight junctions; TLR4, toll-like receptor 4, NF-κB, nuclear factor kappa-light-chain-enhancer of activated B cells; IL-6, interleukin 6; TNF-α, tumor necrosis factor-alpha; RNS, reactive nitrogen species; ROS, reactive oxygen species; ER, endoplasmic reticulum; SFA, saturated fatty acids.

**Table 2 cells-10-03164-t002:** The impact of a high-fat diet (HFD) on gut microbiota.

Bacteria	HFD Impact
**Phylum: *Firmicutes***	↑
**Order**: *Erysipelotrichales*	↑
Class: Bacili	↑
Genus: *Lactobacillus*	↑
**Order**: *Clostridiales*	↑
Genus: *Oscillibacter*	↑
Genus: *Dorea*	↑
Genus: *Ruminococcus*	↑
**Phylum: *Bacteroidetes***	↓
Family: *Prevotellaceae*	↓
Family: *Rikenellaceae*	↓
**Phylum: *Actinobacteria***	↑/↓
Genus: *Bifidobacterium*	↑/↓
**Phylum: Tenericutes**	↓
**Phylum: *Proteobacteria***	↑
**Order**: *Enterobacteriales*	↑
Family: *Enterobacteriaceae*	↑
**Order**: *Desulfovibrionales*	↑
Genus: *Desulfovibrio*	↑
Species: *Bilophila wadsworthia*	↑
**Phylum: *Verrucomicrobia***

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
