# Peer review of "High-Fat, Western-Style Diet, Systemic Inflammation, and Gut Microbiota: A Narrative Review"

_cells, 2021, doi:10.3390/cells10113164_

Round 1

Reviewer 1 Report

cells-1434668

This is a well-written and comprehensive reviews, one of the many on microbiota and CHD. This one focuses on high-fat Western diets, which cause inflammaging (see the highly cited Calder review in Ageing Res Review, there are plenty of topics to be discussed there).

Some issues are missing and have been discussed in an MDPI journal: PMID: 32492487. One, for example, is that of emulsifiers, which often accompany fatty foods and discretionary foods. In the same review the authors discuss one of the most important limitations of microbiota research, i.e. the definition of eubiosis (see JAMA, 321 (7) (2019), pp. 633-635.

Finally, artificial sweeteners are also hypothesized to play a role in atherosclerosis via microbiota modulation. This is still up in the air, but PMID: 32492487 and references therein thoroughly discuss this issue.

Another useful paper to read and discuss is PMID: 30655101

There is a misspelling in Fig 1 (inflamation).

In short: discuss the aforementioned papers and some of their references, down tone some statements (the whole microbiota field is still very hypothetical and there are no authorized health claims), further edit and proof the paper.

Author Response

Response to Reviewer 1 Comments

This is a well-written and comprehensive reviews, one of the many on microbiota and CHD. This one focuses on high-fat Western diets, which cause inflammation (see the highly cited Calder review in Ageing Res Review, there are plenty of topics to be discussed there).

Point 1: Some issues are missing and have been discussed in an MDPI journal: PMID: 32492487. One, for example, is that of emulsifiers, which often accompany fatty foods and discretionary foods. In the same review the authors discuss one of the most important limitations of microbiota research, i.e. the definition of eubiosis (see JAMA, 321 (7) (2019), pp. 633-635.

Response 1: We would like to thank the Reviewer for the thoughtful comments. The corresponding paragraphs regarding emulsifiers, microbiota research limitations, and eubiosis have been added to the introduction, 3.1 and 3.3  sections, respectively.

Point 2: Finally, artificial sweeteners are also hypothesized to play a role in atherosclerosis via microbiota modulation.

Response 2: We would like to thank the Reviewer for the comment. Undoubtedly, sweeteners contribute to the adverse effects of the Western diet. However, this aspect goes beyond the scope of our review, which focuses on fats. All the same, a proper comment has been added to the manuscript.

Point 3: This is still up in the air, but PMID: 32492487 and references therein thoroughly discuss this issue. Another useful paper to read and discuss is PMID: 30655101

Response 3: We referred to the Reviewer’s suggestions. However,  JAMA paper<321 (7) (2019), pp. 633-635>, although very interesting, concerns probiotics research, which is not the focus of our review.

Point 4: There is a misspelling in Fig 1 (inflammation).

Response 4: The misspelling was corrected.

Point 5: In short: discuss the aforementioned papers and some of their references, down tone some statements (the whole microbiota field is still very hypothetical and there are no authorized health claims), further edit and proof the paper.

Response 5: The aforementioned papers and some of their references were discussed. The proper note: “that the dysbiosis field has still more questions than answers, and there are just a few authorized health claims regarding restoring the physiological eubiotic balance” was added in paragraph 3. The paper was edited and checked by the professional proofreading service.

Reviewer 2 Report

This is an interesting, comprehensive, and well-presented review article. The authors have included basic and clinical information to help the readers understant the complexity between high-fat feeding, and gut pathology that leads to systemic inflammation and disease. Please correct a typo in page 790 (add brackets to the references cited). 

Author Response

Response to Reviewer 3 Comments

This is an interesting, comprehensive, and well-presented review article. The authors have included basic and clinical information to help the readers understand the complexity between high-fat feeding, and gut pathology that leads to systemic inflammation and disease.

Point 1: Please correct a typo in page 790 (add brackets to the references cited). 

Response 1:The relevant corrections have been incorporated into the text.

Reviewer 3 Report

- Abstract: Some more details on the general function of gut microbiota should be briefly introducted.

- Table 1: A lot of space is ‘wasted’ here. The table can be displayed much shorter.

- 2. Methodology, line 112-114: The authors state that they excluded articles concerning the influence of single food products or substances on microbiome or inflammation. At least as an example, such papers could be mentioned.

- Line 119: 1013 should be 1013.

- Sometimes HDF is used instead of HFD, please carefully check.

- Line 670: References information: Brackets are missing.

Author Response

Response to Reviewer 3 Comments

Point 1: Abstract: Some more details on the general function of gut microbiota should be briefly introducted.

Response 1:We would like to thank the reviewer for her/his careful reading and valuable comments. We added a short note about the microbiota function as suggested.

Point 2: - Table 1: A lot of space is ‘wasted’ here. The table can be displayed much shorter.

Response 2:The whole table was revised according to the suggestion.

Point 3: Methodology, line 112-114: The authors state that they excluded articles concerning the influence of single food products or substances on microbiome or inflammation. At least as an example, such papers could be mentioned.

Response 3:The effect of a single food product or substance on microbiota and inflammation is an exciting issue. Such a topic could create the basis for a separate systematic review.

Point 4: - Line 119: 1013 should be 1013.

Response 4:The mistake was corrected

Point 5:- Sometimes HDF is used instead of HFD, please carefully check.

Response 5:The manuscript was corrected.

Point 6:- Line 670: References information: Brackets are missing.

Response 6: The missing brackets were added.

Round 2

Reviewer 1 Report

Nice revision